# Deep learning based polyp image comparisons

Anonymous Full Paper
Submission 55

## Abstract

A key challenge in colon capsule endoscopy (CCE) analysis is accurately identifying whether multiple images depict the same polyp, which is critical to avoid overcounting and unnecessary clinical interventions. Additionally, this prevents prolonged follow-up colonoscopies where endoscopists search for duplicate polyps without precise localization. This study presents a novel contrastive learning-based approach to polyp matching using a Siamese neural network architecture. The model trains on a dataset from the Danish CareForColon2015 trial. The dataset comprises 5736 image pairs of confirmed same and different polyps, annotated by experienced clinicians. The proposed model employs ResNet variants (ResNet18, ResNet34, ResNet50) as backbones and utilizes contrastive learning with cosine and Euclidean distance metrics for classification. Experimental results demonstrate high classification accuracy, with optimal performance achieved using a cosine distance threshold of 0.15, yielding an accuracy of 92.0% and an AUC of 0.98.

## 1 Introduction

Colon capsule endoscopy (CCE) is increasingly employed for imaging and inspection of the colon [1, 2]. In a CCE procedure, the patient swallows a capsule camera, which then traverses the gastrointestinal tract and exits via the anus. Prior to CCE, the patient goes through a procedure to clean their intestines [3]. There are several colon camera pills on the market that are quite similar [4]. Data of the current study are derived from CCE procedures conducted using second generation colon capsules (PillCam™ Colon 2, Medtronic, USA). This pill is 31.5 mm long and has a diameter of 11.6 mm. Each end of the pill is equipped with a camera, and each camera head can function independently of the other. Together, the two camera heads achieve a maximum frame rate of 35 frames per second [5]. Typically, each camera head captures 10,000 images.

If one or more polyps are identified in a CCE video, their removal may be deemed necessary [6]. However, if the clinician is inaccurately informed of a higher number of polyps than actually present, additional time may be spent searching for non-existent polyps. Additionally, since the number of polyps identified is part of the risk allocation the patients receive (i.e. more than 2 polyps = moderate risk as opposed to low), double counting of polyps may ultimately result in unnecessary follow-up, causing additional costs and discomfort or anxiety to patients.

In a clinical setting, the camera capsule is known to exhibit bidirectional movement, occasionally traveling from the rectum back towards the proximal colon. Consequently, polyps and other features may be visualized multiple times during a CCE investigation. To prevent the same polyp from being counted more than once, it is essential to have a reliable method for determining whether two or more images of polyps correspond to the same polyp.

There are several factors that make polyp matching a challenging task. First, a single polyp can appear significantly different when viewed from varying angles or distances. Second, the capsule's onboard lighting can create diverse illumination conditions, including specular reflections. Third, the cleanliness of the colon can change over time affecting image consistency.

When performing polyp matching, other contextual factors—such as the time interval between images, the estimated location within the colon, and the size and type of the polyps—can also be considered.

In this paper, we focus on a fundamental question: "Given two images, both assumed to contain a polyp, are they showing the same polyp?". To the best of our knowledge, this is the first attempt towards addressing this research question.

## 2 Methodology

This section introduces the dataset and approach of this study.

### 2.1 Dataset and pre-processing

In order to address the research question, we need at least two images known to be of the same polyp. Ideally, one would like to have images separated in time of the same polyp and seen from multiple directions. However, no such curated public dataset exists. In this study, we use the Danish CareForColon2015 trial [7, 8] as the main data source. From this dataset, clinicians from HOSPITALNAME University Hospital (including authors X and Y) have for a large set of polyps exported up to five images of the same polyp from the same passing. The exported images show 1) First partial, 2) First full,

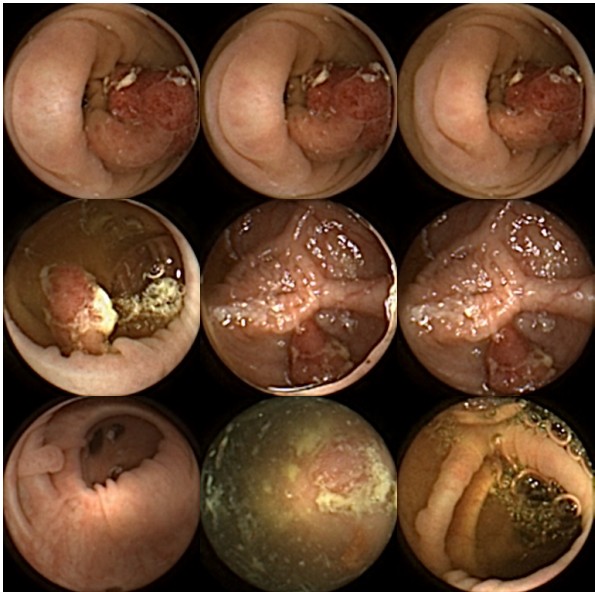

**Figure 1.** Top row: Example images of the same polyp (Easy). Middle row: Example images of the same polyp (Hard). Bottom row: Example images of three different polyps. Figure also shows varying degree of cleansing quality and artifacts.

3) Best full, 4) Last full, 5) Last partial, image of the same polyp. This does not give us images of the same polyp separated significantly in time. However, for each polyp we can use three pairs (2-3, 2-4, 3-4) of images showing the same full polyp. We use this as training data for the "same polyp" class. For the "different polyps" class, random sets of images from different polyps are chosen.

We have for the current study access to 2780 polyps from 853 patients (The full dataset is constantly growing and currently consists of about 6000 polyps). From this data set, we only use polyps that have been assigned all five images (partial, full etc), to be sure that we are working with images that show the whole polyp. This results in 1912 polyps from 754 patients. From 1912 polyps, each with three pairs, we get 5736 image pairs of the same polyp. From the large number of pairs of different polyp images, we draw a random subset of 5736 pairs. As the same polyp might wrongly be labeled as different polyps for a given patient, we do not use images from assumed different polyps from the same patient. In the end, we have a balanced training set of 5736 samples for each of the two classes i.e., same polyp and different polyp.

Figure 1 shows examples of images pairs of the same polyp (upper two rows) and of different polyps (lower row). The cleaning quality and presence of artifacts (like bubbles) can vary. We can note that depending on the angle or distance and the presence of artifacts in different images of the same polyp, the task of matching polyps can become quite difficult.

## 2.2 Network architecture

In this section, we discuss the suggested Siamese deep neural network for comparing images.

To assess whether two images, both containing polyps, are of the same polyp, we train a Siamese deep neural network based on contrastive learning [9–11].

A Siamese network [12] consists of two identical networks that have the same structure and shared weights. When doing inference, two separate images are fed through each of these two "arms" of the Siamese network. The outputs of the two networks are then fed into an embedding layer of size 128, which results in a final embedding vector of length 128 for each image. The distance between these two embedding vectors forms the basis of the final classification, where a large distance is an indication of the images not being of the same polyp.

In this study, we employ the cosine and Euclidean vector distance for measuring distances. A pair is classified as the same polyp if the distance is below a preset threshold.

We use the ResNet family [13] of deep neural nets as the backbones of the Siamese network. ResNets come in different sizes, where the size parameter assigns how many residual blocks the full net consists of. In this study we use ResNet18, ResNet34 and ResNet50 as the backbone.

## 2.3 Training

In this section, we discuss the training approaches and parameters of our experiments. The input images are resized to 224x224 pixels to fit the input dimensions of the ResNets. The images are scaled to have zero mean and 1 standard deviation for each color channel. We use on-the-fly data augmentation techniques [14] to increase the effective sample size. For this, we use random rotation (all angles), random horizontal and vertical flips (p=0.5), and color jittering (contrast=brightness=0.2).

We randomly split the polyps of the dataset 0.7/0.15/0.15 into training, validation and test sets. Images from a given polyp are in only one of these sets, but different polyps from a given patient can be in different sets.

For this study, we use a contrastive loss function [15], as it attempts to push images of different polyps further apart, while images of the same polyp should have as low as possible vector distance [16]. This loss is defined as

$$L_C = (1 - Y) \cdot D_W^2 + Y \cdot \max(0, m - D_W)^2, \quad (1)$$

where $Y$ is 0/1 for same/different image pairs, $D_W$ is the distance measure between the two embedding vectors, and $m$ is a margin. This study uses 0.5 and 1 as margins for cosine and Euclidean distances, respectively.

We use a batch size of 32, and the Adam optimizer with a fixed learning rate of 0.001.

We run the training for up to 100 epochs and use early stopping with a patience value of 10 and minimum loss delta of 0.001 to avoid overfitting. This means that if a model trains for 40 epochs, the model obtained at 30 epochs is chosen as the best model.

The network is trained on a NVIDIA RTX 2000 Ada generation graphics card, and the runtime was in the range of a few hours. No structured hyperparameter optimization is performed. We analyze the impact of using different vector distance thresholds on the performance. For our experiments, other backbone networks or loss functions are not considered.

To classify a pair of polyp images as being of the same polyp, the vector distance of the embedding vectors must be below a selected threshold. For our initial experiments, we use 0.5 and 0.3 as thresholds for the Euclidean and cosine distances, respectively.

## 3 Results

We first chose to train a ResNet34. It was trained separately for both cosine and Euclidean vector distances. The training and validation loss curves for the two metrics can be seen in Figure 2–both training sessions stop early. The best models are marked with a red dot in the figure. After training the Siamese network, its performance is tested on an independent test set. Using the classification cut-offs of 0.5 and 0.3 for Euclidean and cosine distance metrics, we achieved an accuracy of 89.6% and 85.7%, respectively. The corresponding confusion matrices are shown in Figure 3.

As the metric thresholds are somewhat arbitrary, they might not be optimal for this task. Figure 4 shows the test set recall per class and accuracy in total as a function of the threshold for the two measures. From this, it seems like the selected threshold for cosine is far from optimal. Following this, we select a new threshold of 0.15 for the cosine distance and retrain the network. This results in a test set accuracy of 92.0%, and the corresponding confusion matrix as shown in Figure 5 displays a more balanced classifier than with the original 0.3 threshold. As this new threshold is based on results from the test set, this new accuracy after retraining with a new threshold can not be said to be on a totally independent test set.

Figure 6 shows the receiver operating characteristics (ROC) curves for the cosine and Euclidean distances as a result of using different thresholds for classifying a pair of polyp images as being of the same polyp. The area under the curve (AUC) of these two curves is 0.98 and 0.96, respectively.

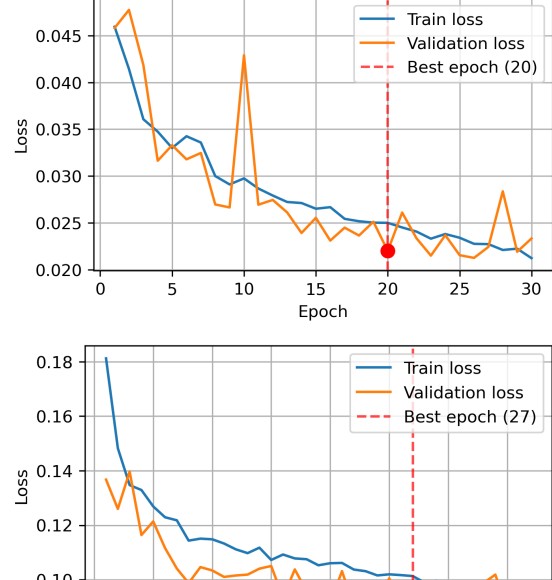

**Figure 2.** ResNet34: Training and validation loss using cosine (upper) and Euclidean (lower) distance metrics. Best model following triggering of early stopping marked in red.

To see if a smaller or larger ResNet is better to use, we also train ResNet18 and ResNet50 models.

Figure 7 shows the training curves for a Siamese network with a ResNet18 as the backbone. Accuracy for the Euclidean and cosine vector distances on the test set was 90.8% and 91.9%, respectively. Figure 8 shows the confusion matrices. Figure 9 shows the optimal threshold for the trained ResNet18 on an independent test set. Finally, Figure 10 shows the ROC curves and AUC values of the ResNet18 models.

Figure 11 shows the training curves for a Siamese network using ResNet50 as the backbone. Accuracy for the Euclidean and cosine vector distances on the test set is 92.2% and 92.0%, respectively. Figure 12 shows the confusion matrices. Figure 13 shows how the optimal threshold looks for the trained ResNet50 on the independent test set. Finally, Figure 14 shows the ROC curves and AUC values of the best ResNet50 models.

The performance on the test set of the different ResNets is summarized in Table 1 for both cosine and Euclidean vector distances. As can be seen in Figures 7-14 and Table 1, both ResNet18 and ResNet50 achieve similar results as ResNet34. The results indicate that the cosine distance metric is slightly better than the Euclidean distance metric for our classification task.

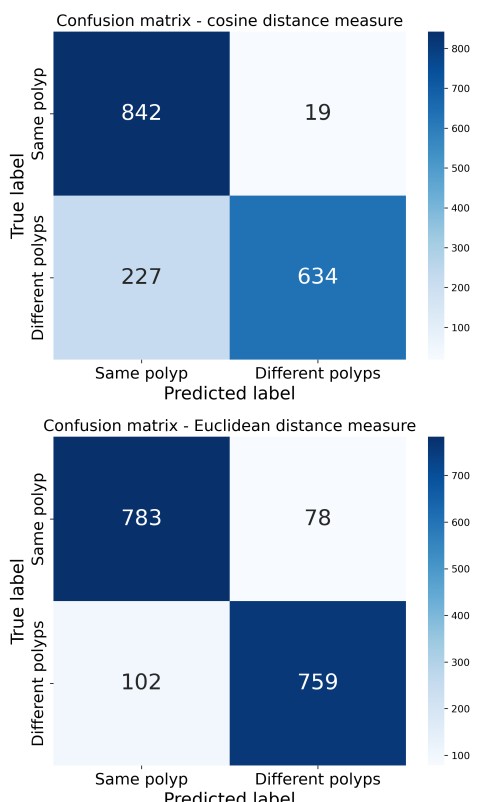

**Figure 3.** ResNet34: Confusion matrices from the independent test set using cosine (upper) and Euclidean (lower) distance metrics.

## 4   Conclusion

This paper presents a deep neural network that takes two colon capsule endoscopy images known to both contain polyps and tries to determine whether they are of the same polyp. Having such an automated tool available when evaluating the results of a CCE investigation would be helpful to avoid over-counting the number of detected polyps. This will both save procedure time when a clinician at a later time goes in and removes polyps and ensure more accurate patient risk allocation and ultimately correct patient follow-up.

Strengths of the study are that we have a large set of images known to be of the same polyp. However, there are some limitations. Most importantly, the presented setup does not include pairs of images known to be of the same polyp, but separated significantly in time. It is also of interest to study how including images that only partially contain a polyp will influence the classification performance. These two aspects will be addressed in future work, where we also will try other training strategies and parameters. It is also of interest to quantify how cleaning/image quality affects classification performance.

The presented approach is first-of-a-kind and shows the potential of using deep neural networks

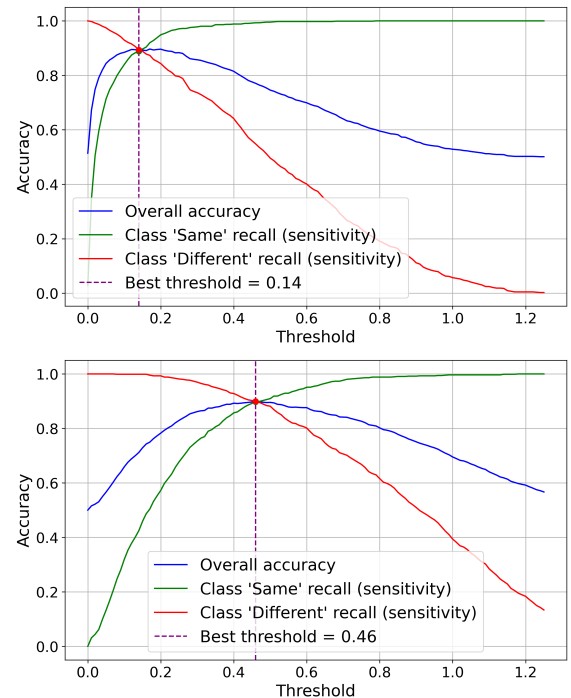

**Figure 4.** ResNet34: Accuracy of independent test as function of threshold set using cosine (upper) and Euclidean (lower) distance metrics.

for polyp matching in colon capsule endoscopy, with the potential to save time and improve care in colon capsule endoscopy.

# Acknowledgments

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

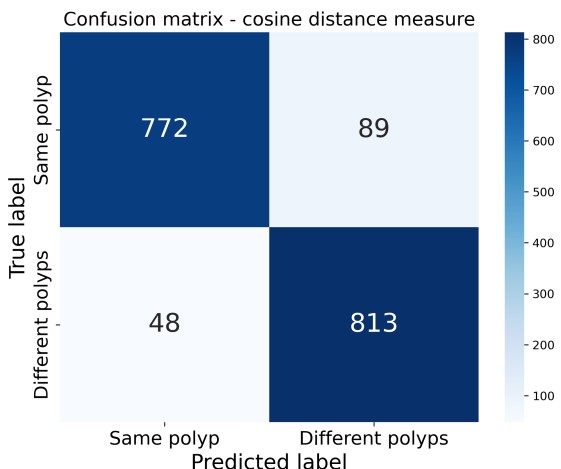

**Figure 5.** ResNet34: Confusion matrices from the independent test set using cosine distance metric with retrained network.

**Table 1.** Performance measures on test set for different ResNets using cosine (upper part) and Euclidean (lower part) vector distance.

| Cosine vector distance | | | |
|---|---|---|---|
| Model | ResNet18 | ResNet34 | ResNet50 |
| Accuracy | 0.9193 | 0.9204 | 0.9199 |
| Precision | 0.8865 | 0.9013 | 0.8934 |
| Recall | 0.9617 | 0.9443 | 0.9535 |
| Specificity | 0.8769 | 0.8966 | 0.8862 |
| F1 score | 0.9226 | 0.9223 | 0.9225 |
| Euclidean vector distance | | | |
| Model | ResNet18 | ResNet34 | ResNet50 |
| Accuracy | 0.9077 | 0.8955 | 0.9001 |
| Precision | 0.9199 | 0.9068 | 0.9206 |
| Recall | 0.8931 | 0.8815 | 0.8757 |
| Specificity | 0.9222 | 0.9094 | 0.9245 |
| F1 score | 0.9063 | 0.8940 | 0.8976 |

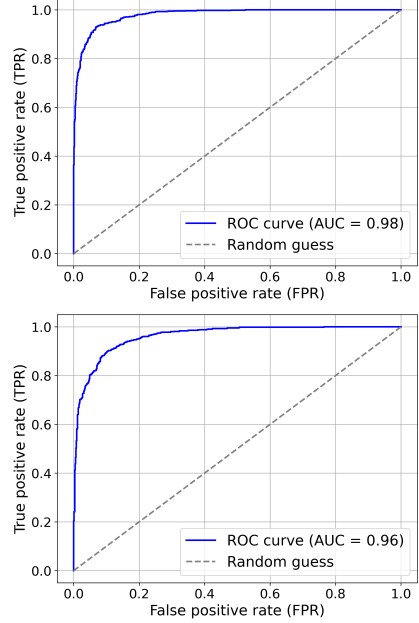

**Figure 6.** ResNet34: Receiver operating characteristics (ROC) curves for the cosine (upper) and Euclidean (lower) distances as a result of using different thresholds for classifying a pair of polyp images as being of the same polyp.

[5] L. Negreanu, R. Babiuc, A. Bengus, and R. Sadagurschi. "PillCam Colon 2 capsule in patients unable or unwilling to undergo colonoscopy". In: *World journal of gastrointestinal endoscopy* 5.11 (2013), pp. 559–567. DOI: 10.4253/wjge.v5.i11.559.

[6] M. Jung. "The 'Difficult' polyp: Pitfalls for endoscopic removal". In: *Digestive Diseases* 30.Suppl. 2 (2012), pp. 74–80. DOI: 10.1159/000341898.

[7] L. Kaalby, U. Deding, M. Kobaek-Larsen, A.-L. V. Havshoi, E. Zimmermann-Nielsen, M. K. Thygesen, R. Kroeijer, T. Bjørsum-Meyer, and G. Baatrup. "Colon capsule endoscopy in colorectal cancer screening: a randomised controlled trial". In: *BMJ Open Gastroenterology* 7.1 (2020), pp. 1–7. DOI: 10.1136/bmjgast-2020-000411.

[8] G. Baatrup, T. Bjørsum-Meyer, L. Kaalby, B. Schelde-Olesen, M. Kobaek-Larsen, A. A. Koulaouzidis, R. Kroijer, I. Al-Najami, N. Buch, A. Høgh, N. Qvist, M. Thygesen, and U. Deding. "Choice of colon capsule or colonoscopy versus default colonoscopy in FIT positive patients in the Danish screening programme: a parallel group randomised controlled trial". In: *Gut* (2025). Published online, to appear in print. DOI: 10.1136/gutjnl-2024-333687.

[9] T. Chen, S. Kornblith, M. Norouzi, and G. Hinton. "A simple framework for contrastive learning of visual representations". In: *International conference on machine learning*. PmLR. 2020, pp. 1597–1607. URL: https://dl.acm.org/doi/pdf/10.5555/3524938.3525087.

[10] O. Ilina, V. Ziyadinov, N. Klenov, and M. Tereshonok. "A survey on symmetrical neural network architectures and applications". In: *Symmetry* 14.7 (2022). DOI: 10.3390/sym14071391.

[11] K. Zhang, S. Qi, J. Cai, D. Zhao, T. Yu, Y. Yue, Y. Yao, and W. Qian. "Content-based image retrieval with a convolutional Siamese neural network: Distinguishing lung cancer and tuberculosis in CT images". In: *Computers*

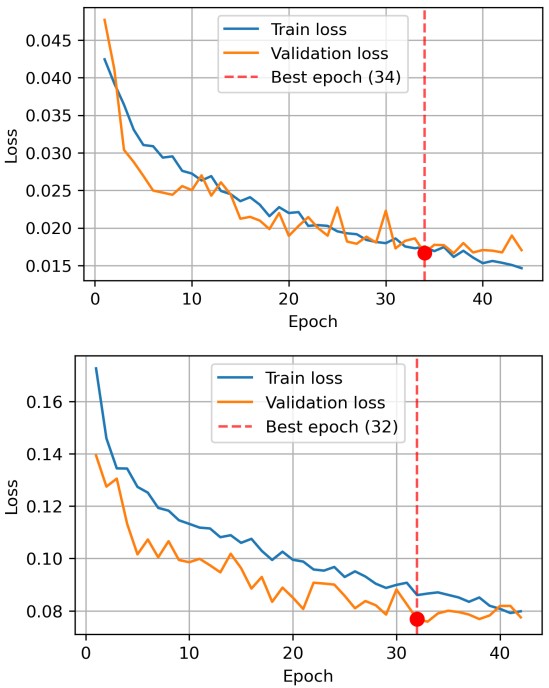

**Figure 7.** ResNet18: Training and validation loss using cosine (upper) and Euclidean (lower) distance metrics. Final model following triggering of early stopping marked in red.

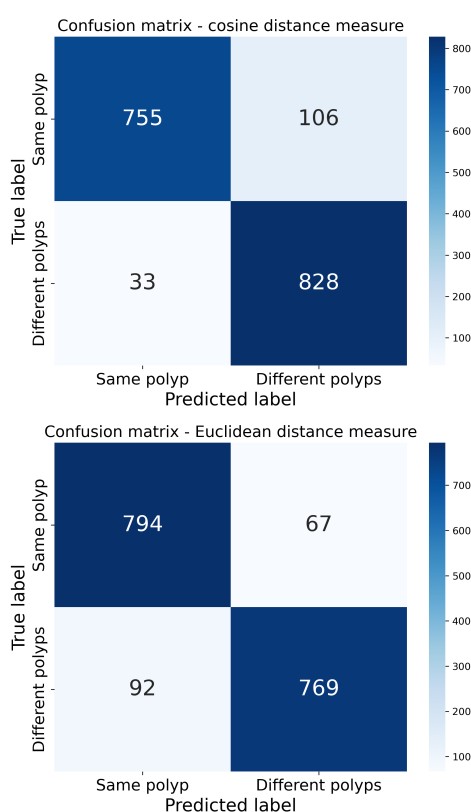

**Figure 8.** ResNet18: Confusion matrices from the independent test set using cosine (upper) and Euclidean (lower) distance metrics.

*in biology and medicine* 140 (2022), pp. 1–10. DOI: 10.1016/j.compbiomed.2021.105096.

[12] J. Bromley, J. Bentz, L. Bottou, I. Guyon, Y. Lecun, C. Moore, E. Sackinger, and R. Shah. "Signature verification using a "Siamese" time delay neural network". In: *International Journal of Pattern Recognition and Artificial Intelligence* 7 (Aug. 1993), p. 25. DOI: 10.1142/S0218001493000339.

[13] K. He, X. Zhang, S. Ren, and J. Sun. "Deep residual learning for image recognition". In: *2016 IEEE Conference on Computer Vision and Pattern Recognition (CVPR)*. 2016, pp. 770–778. DOI: 10.1109/CVPR.2016.90.

[14] S. Yang, W. Xiao, M. Zhang, S. Guo, J. Zhao, and F. Shen. *Image data augmentation for deep learning: A survey*. 2023. URL: https://arxiv.org/abs/2204.08610.

[15] R. Hadsell, S. Chopra, and Y. LeCun. "Dimensionality reduction by learning an invariant mapping". In: *2006 IEEE Computer Society Conference on Computer Vision and Pattern Recognition (CVPR'06)*. Vol. 2. 2006, pp. 1735–1742. DOI: 10.1109/CVPR.2006.100.

[16] T. Xiao, X. Wang, A. A. Efros, and T. Darrell. "What should not be contrastive in contrastive learning". In: *CoRR* abs/2008.05659 (2020). URL: https://arxiv.org/abs/2008.05659.

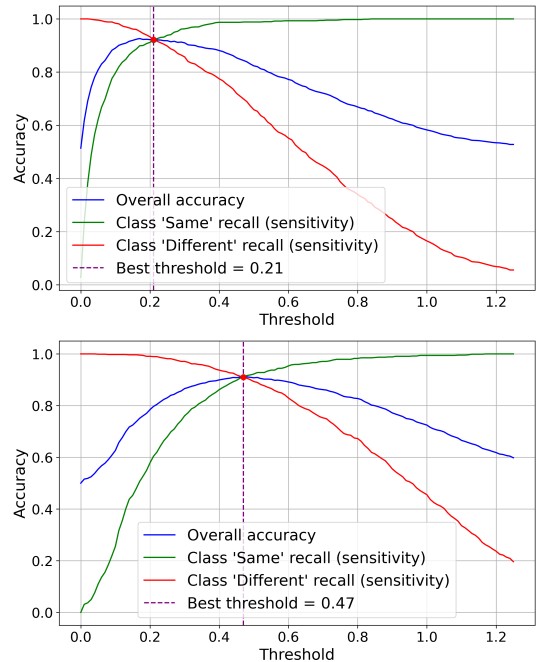

**Figure 9.** ResNet18: Accuracy of independent test as function of threshold set using cosine (upper) and Euclidean (lower) distance metrics.

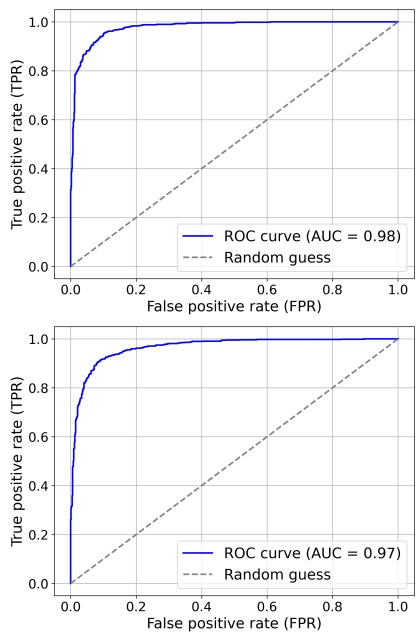

**Figure 10.** ResNet18: Receiver operating characteristics (ROC) curves for the cosine (upper) and Euclidean (lower) distances as a result of using different thresholds for classifying a pair of polyp images as being of the same polyp.

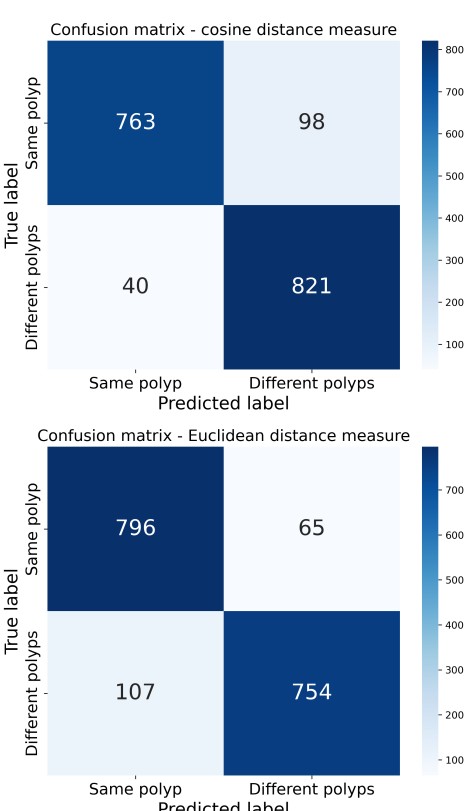

**Figure 12.** ResNet50: Confusion matrices from the independent test set using cosine (upper) and Euclidean (lower) distance metrics.

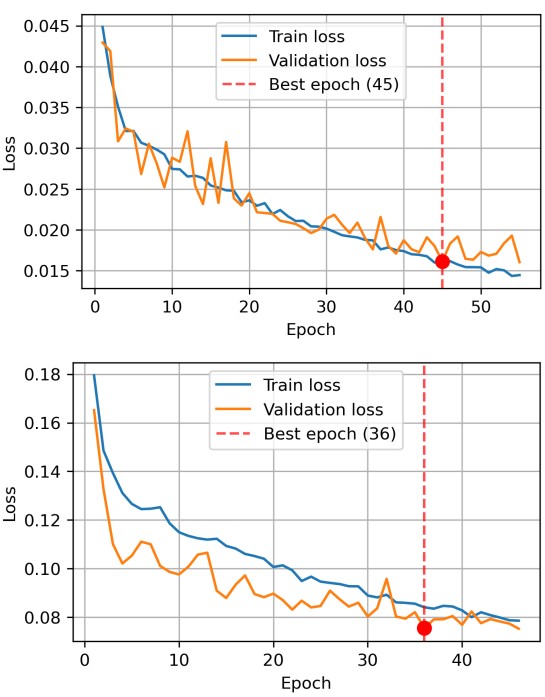

**Figure 11.** ResNet50: Training and validation loss using cosine (upper) and Euclidean (lower) distance metrics. Final model following triggering of early stopping marked in red.

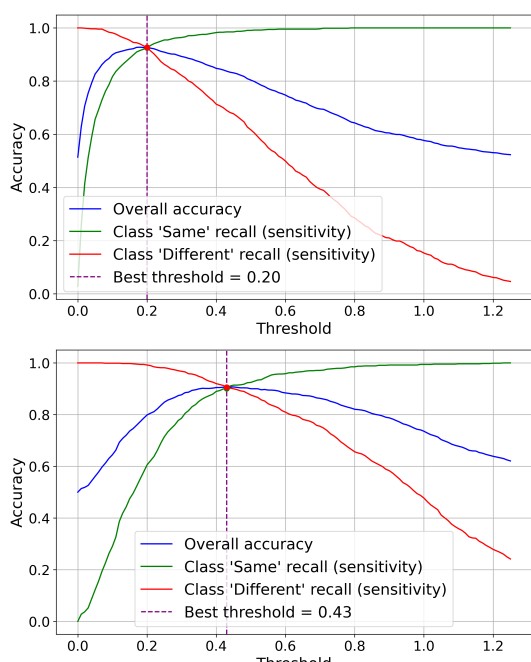

**Figure 13.** ResNet50: Accuracy of independent test as function of threshold set using cosine (upper) and Euclidean (lower) distance metrics.

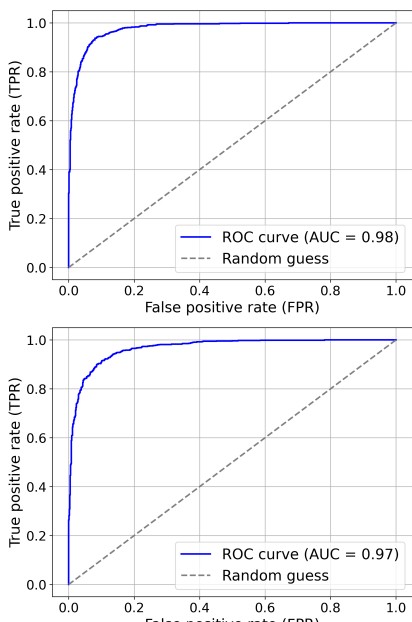

**Figure 14.** ResNet50: Receiver operating characteristics (ROC) curves for the cosine (upper) and Euclidean (lower) distances as a result of using different thresholds for classifying a pair of polyp images as being of the same polyp.

