# OpenReview forum: "Deep learning based polyp image comparisons"
_NLDL.org/2026/Conference — Submitted to NLDL 2026_

### Official Review · Reviewer_Rzru · 2025-09-22
**Good paper with contributions, but needs stronger benchmarking**

**Rating:** 4
**Confidence:** 4
**Final Rating:** 4
**Final Confidence:** 3

**Summary:**

This paper addresses a clinically important and previously unexplored problem in colon capsule endoscopy, determining whether two images correspond to the same polyp. This is crucial to avoid double counting, unnecessary follow-ups, and misclassification of patient risk. The authors propose a contrastive learning approach with a Siamese neural network using ResNet backbones.

The dataset comprises pairs of same-polyp images and an equal number of different-polyp pairs, all annotated by clinicians. Results show high accuracy (up to 92.2%) and excellent AUC values (up to 0.98), suggesting the feasibility of deep learning for automated polyp matching. The paper is clearly written, presents the methodology systematically, and discusses both strengths and limitations.

**Strengths:**

The paper is easy to follow, with clear figures and a transparent discussion of limitations and future directions. This seems to be the first work tackling the task of polyp matching across images in CCE, which is highly relevant for clinical practice. The proposed approach, if validated further, could reduce double counting of polyps, save clinician time, and improve patient risk stratification.

The study leverages a large dataset of carefully selected image pairs from a real clinical trial, ensuring some medical relevance.

The choice of Siamese networks with contrastive loss is well-motivated and appropriate for similarity learning problems.

Achieving >90% accuracy and AUC close to 1.0 across different ResNet backbones demonstrates the technical potential of the method.

**Weaknesses:**

-The paper does not compare against established baselines or state-of-the-art (SOTA) methods for either this dataset or related medical image retrieval tasks. This makes it difficult to assess how much of the performance gain comes from the proposed approach versus standard similarity learning techniques.

-The adjustment of the cosine distance threshold based on test set results weakens the independence of the evaluation, and may inflate reported performance. A validation-driven or cross-validation strategy would be more rigorous.

-All results are reported as single runs, without averages or variances across different seeds. Given the stochastic nature of deep learning training, this is an important omission for evaluating robustness.

-Same-polyp images are taken within the same capsule passage, not across different temporal segments, which limits the generalization potential.

-While the authors acknowledge limitations, the conclusion could better reflect that results demonstrate feasibility rather than readiness for clinical deployment.

Questions to authors:

1) Why did you not include a comparison to SOTA methods in medical image similarity or retrieval, even if not specifically designed for polyps?

2) Can you report mean ± standard deviation over multiple training runs to strengthen the reliability of your results?

3) How might your model perform if evaluated on polyps imaged at significantly different times (e.g., different capsule passes or procedures)?

**Final Justification:**

The research is relevant, and the proposed approach is interesting. However, the lack of a suitable benchmark leads me to suggest a weak acceptance. The authors state that a state-of-the-art benchmark is outside the scope of the proposed work, but they haven't provided a justification that I can understand as adequate. This could be a factor in its rejection in the event of a tie with other papers.

**Justification:**

This paper is novel, relevant, and technically correct, and it introduces a clinically significant task with a promising solution. The methodology is appropriate, the dataset is medically meaningful, and the reported performance is great. However, the lack of benchmarking against established methods, the use of the test set for threshold tuning, and the absence of multiple-run statistics weaken the strength of the evidence. Furthermore, the dataset setup limits the demonstrated generalizability of the approach. Despite these weaknesses, the contribution is valuable and opens a promising research direction. With more modest claims and additional experiments (baselines, robustness analyses, etc), this work could become a great paper.

---

> ### Author Rebuttal · Authors · 2025-10-22
>
> # Comparison against other similarity learning techniques/medical image retrieval tasks
> Another reviewer asked something similar where he/she asked about comparing against MoCov2 and SimCLR. As we stated there, we feel that this is outside the scope of this paper, but that this can be done in a future more comprehensive paper.
>
> # Re-training based on test set results
> This is a point that has also been raised by other reviewers. The intention of our choice was to illustrate that different parameter choices influence the test set performance metrics, but we agree that this was done in a bad way. We will in a revised version of the paper train the networks as before, and then for each trained network run a grid search using the validation set only to produce a threshold value that produce a balanced confusion matrix. This threshold value will then be used to produce the final performance metrics of the untouched test set.
>
> # Single run results only
> Yes, we could have set up a cross-validation type of training scheme that would produce average performance values and their variances. This has been implemented and the results of 10-fold cross-validation are reported below for the Euclidean distance (we have not had time to run the cosine distance).
>
> # Lack of time-separation for same-polyp images
> Yes, this is a valid comment. In a dream situation, we would have many polyps that are known to be seen with a large time-separation (i.e. the camera pill traveled backwards to an earlier seen polyp). The problem is that it is extremely time-consuming to produce such data sets, especially if one wants a high quality/low chance of mislabeling. We do not have a number on how often this happens in a patient, but probably for less than 10% of polyps. To produce this dataset, a clinician must carefully look through all images and try to determine where the camera pill is in the colon. And when a “new” polyp is seen, the clinician must make an educated guess on whether this polyp was seen before or not. It is further complicated by the fact that there is no automated way of knowing where (even roughly) in the colon the pill camera is at a given time.
> So, generating high-quality data on polyps seen with a large time separation would take weeks only to get a small number like 100. When tools for estimating the camera pill position in the colon are available in the future, producing such datasets of a reasonable size might be feasible.
>
> # Feasibility study
> Yes, we could have stressed that it is a feasibility/proof-of-concept study where we have not tried to do a complete hyperparameter optimization. This re-training approach (which reviewers have pointed out is not good practice) was also not meant as producing a ready-to-use model, but to show that one might be able to increase the performance via hyperparameter optimization. Based on the comments, the revised version will be adjusted to reflect that.
>
> # Questions to authors (bullet points are answers):
> 1. Why did you not include a comparison to SOTA methods in medical image similarity or retrieval, even if not specifically designed for polyps?
> * As mentioned above, we think this is outside the scope of this paper, but this is something we will look into in the future.
> 2. Can you report mean ± standard deviation over multiple training runs to strengthen the reliability of your results?
> * Yes, we have run a 10-fold cross-validation. Each time we selected 10% of the datapoints to be in the test set (each data point is in only one of the 10 test sets) and used the remaining 90% of the data for training (80% of the 90%) and validation (20% of the 90%).
> * As this training does take some time, we here in the rebuttal only report the results of using the Euclidean distance using the ResNet34 (but the other ResNets and cosine distance will be in a revised version of th epaper). The results given here are from 10 results based on early stopping with a patience of 10, but we are considering using 20 as patience for the revised paper (following comments from a reviewer).
> * The mean accuracy was 0.9012 with a standard deviation of 0.0081.
> 3. How might your model perform if evaluated polyps imaged at significantly different times (e.g., different capsule passes or procedures)?
> * Within the same procedure, the degree of colon cleansing quality could be something that can be expected to vary the most over time. While the appearance of a polyp is not expected to vary much. The data set does contain images of a given polyp from both directions or views, if present. Without a reasonably large set of cases with the same polyp seen twice with clear time separation, one cannot know how the performance would be.

---

### Official Review · Reviewer_3ZVX · 2025-09-25
**Deep learning based polyp image comparisons**

**Rating:** 2
**Confidence:** 5
**Final Rating:** 2
**Final Confidence:** 3

**Summary:**

This paper proposes a method for polyp re-identification through image comparison. The authors employ a contrastive learning approach that uses paired polyp images, where the model learns to bring similar polyps closer together in the feature space while pushing dissimilar polyps further apart.

At inference time, the model is presented with two polyp images, and if the computed distance is below a predetermined threshold, the polyps are classified as being the same.

**Strengths:**

1. This paper addresses an important problem in colon capsule endoscopy.
2. The paper is well-written overall, with the importance and challenges of the task clearly presented.
3. The experimental design is well-structured.

**Weaknesses:**

My primary concerns relate to the evaluation methodology, dataset description, and hyperparameter optimization procedures.

Dataset description:

1. The dataset description between lines 103 and 119 lacks clarity and requires rewriting for better comprehension. Lines 110-113 appear to be repetitive.

2. In section "2.3 Training" (line 166), the authors state that polyps from the same patient can appear in different dataset splits. However, in section "2.1 Dataset and pre-processing" (line 114), they indicate that different polyps from the same patient are not used. This contradiction requires clarification.

3. Including images from the same patient in different dataset splits constitutes data leakage, which is problematic for reliable evaluation. In polyp detection tasks, training on images that are closely related to test images is considered bad practice.

4. Are the "different polyps" class pairs constructed by combining a polyp from one patient with a randomly selected polyp from another patient? If so, this presents an evaluation bias, as inter-patient variability may be substantially higher than intra-patient variability. The accuracy for distinguishing different polyps from the same patient should be assessed, as this represents the clinically relevant task.

Hyperparameter optimization:

The authors recompute threshold values using the test dataset, which is considered bad practice in machine learning principles. Only results obtained using thresholds determined exclusively from training/validation data should be reported.

Questions:

1. Why are polyp pairs 1) and 5) not utilized as additional data augmentation variants / positive pairs?

2. Why is network retraining performed with the newly computed threshold (line 223)? This threshold should be a test-time parameter that does not require model retraining, since it is not involved in the training objective (Equation 1).

3. Why not sample negative polyp pairs randomly from each mini-batch instead of relying on fixed pairs throughout the entire training process?

4. What is the purpose of polyp categories 1) and 5) defined in lines 94-95? Based on my understanding, only categories 2, 3, and 4 are utilized in the study.

**Final Justification:**

I am generally satisfied with the authors' explanations addressing my questions and comments.

If the authors implement the various improvements discussed by the other reviewers, this work could serve as a solid proof-of-concept for a future submission.

Given that the required updates represent substantial changes to the manuscript, I believe it should be reviewed again with the revised version to ensure proper implementation of the discussed improvements.

**Justification:**

While the research topic is highly relevant and the experimental approach shows merit, the methodological concerns significantly outweigh the contributions. Specifically, the optimization of hyperparameters on the test set and the presence of data leakage through patient overlap between training and test sets does not respect best practices in both the AI/ML and medical communities.

I recommend that the authors recompute all results without fitting thresholds on the test set and ensure complete separation of patients between training and test splits.

---

> ### Author Rebuttal · Authors · 2025-10-22
>
> # Dataset description
>
> ## Rewriting for better comprehension
> The reviewer states that lines 100-113 appear to be repetitive. The lines are
> “From 1912 polyps, each with three pairs, we get 5736 image pairs of the same polyp. From the large number of pairs of different polyp images, we draw a random subset of 5736 pairs”. We the authors fail to see the repetition here. The first sentence is about the pairs of images of the same polyp, while the second sentence is about the pairs of images of different polyps. Regarding rewriting, maybe we should have explained where this “three pairs” comes from, and pointed back to the paragraph before this. Maybe we should have spent more time on explaining what we mean by “As the same polyp might wrongly be labeled as different polyps”. This comes from when the clinical experts have looked through the videos, where they sometimes will label the revisiting of an already observed polyp as a new polyp. So, by not using polyp images of the same patient, we make sure to not have a pair of images that we think are from different polyps, but are actually from the same polyp. This is connected to the comments/answers further down.
>
> ## Polyps and patients and splits
> The reviewer writes: "In section “2.3 Training” (line 166), the authors state that polyps from the same patient can appear in different dataset splits. However, in section “2.1 Dataset and pre-processing” (line 114), they indicate that different polyps from the same patient are not used. This contradiction requires clarification."
> This could have been written clearer. For images of a given polyp, it is either in training, validation, or test set. However, since many patients have multiple polyps, it is possible that a patient’s polyps might contribute polyp images to more than one of the training, validation, or test sets. This is about the main split into training, validation, and test sets. In section 2.1, we talk about how the pairs are constructed, and here we enforce the requirement that a pair of different polyp images cannot contain images from the same patient.
>
> ## Data leakage
> The reviewer writes: "Including images from the same patient in different dataset splits constitutes data leakage, which is problematic for reliable evaluation. In polyp detection tasks, training on images that are closely related to test images is considered bad practice."
> This and the next comment and answer are connected. We did think through these issues, and our initial thought was that this did look like a problematic data leakage. We could have chosen to include all polyps of a given patient in only one of training, validation or test set. As the human colon only to a small degree has meaningful visual differences across patients, this is not expected to be a real problem. Furthermore, the network is trained to detect differences/similarities between two images, and not to detect a polyp or other artifacts in the images. So even if the model saw images of polyp1 of patient X during training, it seems highly unlikely that this is any way would be helpful for the network to “cheat” when comparing images of polyp2 of patient X either with other polyp2 images or with images of a polyp of patient Y.
>
> However, we have rewritten the code so that all polyps of a given patient are strictly in only one of the training, validation, or test sets. Or initial retraining using these sets does not indicate any significant performance differences compared to what is reported in the initial submission.
>
> ## Clinical relevance of selecting polyp pairs
> The reviewer writes: "Are the “different polyps” class pairs constructed by combining a polyp from one patient with a randomly selected polyp from another patient? If so, this presents an evaluation bias, as inter-patient variability may be substantially higher than intra-patient variability. The accuracy for distinguishing different polyps from the same patient should be assessed, as this represents the clinically relevant task."
> This and the comment above are connected. The reviewers comment is correct and shows a deep understanding of the problem. Ideally, as the reviewer writes, the task should be to compare only images within the same patient (as this is the only clinically relevant task). There are several reasons for why this is not done. Firstly, as mentioned there will be cases where the expert clinician thinks images actually of the same polyp are from different polyps. Then we risk introducing mislabeled data to the training set. Secondly, this would restrict the dataset to patients that have more than one polyp. We feared that one then would potentially not adequately include smaller polyps (as persons with a single polyp are assumed to on average have a smaller polyp than persons with several polyps). One might claim that the only clinically relevant setting is actually of patients with more than one polyp. However, this is the task of the polyp matching to determine.
>
> The reviewer speculates that “inter-patient variability may be substantially higher than intra-patient variability”. It is not known how large this difference is, but clinicians in our group suspect that human colons look very similar across patients.
>
> However, we can redo the training using only images from within the same patient to construct “different polyp” pairs, if this is seen as needed for a revised manuscript.
>
> # Hyperparameter optimization
>
> ## Re-training based on test set results
> The authors recompute threshold values using the test dataset, which is considered bad practice in machine learning principles. Only results obtained using thresholds determined exclusively from training/validation data should be reported.
> This is a point that has also been raised by other reviewers. The intention of our choice was to illustrate that different parameter choices influence the test set performance metrics, but we agree that this was done in a bad way. We will in a revised version of the paper train the networks as before, and then for each trained network run a grid search using the validation set only to produce a threshold value that produce a balanced confusion matrix. This threshold value will then be used to produce the final performance metrics of the untouched test set.
>
> ## Questions (bullet points are answers)
> 1. Why are polyp pairs 1) and 5) not utilized as additional data augmentation variants / positive pairs?
> * The images “1) First partial” and “5 Last partial” will often contain a very small part of the polyp and maybe seen far away in the image. These types of images would not make sense to be considered for polyp matching.
> 2. Why is network retraining performed with the newly computed threshold (line 223)? This threshold should be a test-time parameter that does not require model retraining, since it is not involved in the training objective (Equation 1).
> * Yes, and this is answered above in your general comment on the hyperparameter optimization.
> 3. Why not sample negative polyp pairs randomly from each mini-batch instead of relying on fixed pairs throughout the entire training process?
> * This could have been done, and in future implementation it will likely be done. However, using fixed data is not uncommon to do. Ref. some studies chose to do data augmentation before training, while some chose to do it on-the-fly.
> 4. What is the purpose of polyp categories 1) and 5) defined in lines 94-95? Based on my understanding, only categories 2, 3, and 4 are utilized in the study.
> * We have described how the clinicians who made this data set worked and what they produced (this data was not made explicitly for this study). When the clinician found a polyp, she or he would step backwards to find the very first image where she or he could see the polyp (which could be a very small fraction of it or far away), which is image 1. Then she would find the first image that showed the whole polyp (image 2). Then she or he would find the image that best shows the full polyp (image 3), and then the last image that shows the full polyp (image 4), and finally the last image with a part of the polyp visible (image 5). Often the clinician would not be able to see all five images, but to avoid having images of just a small part of a polyp, we restricted the study to those polyps that had images 1-5, and then we only used image 2-4. Another reviewer suggested that we should have shown example images of the partial images (1 and 5), and we agree that this would have made things clearer. We will do that in the revised version of the paper.

---

### Official Review · Reviewer_9nxB · 2025-09-26
**Siamese Network for PolyP Matching in Colon Capsule Endoscopy**

**Rating:** 2
**Confidence:** 2
**Final Rating:** 2
**Final Confidence:** 2

**Summary:**

The paper tackles a key challenge in colon capsule endoscopy (CCE) analysis: identifying whether different images correspond to the same polyp. Reliable polyp matching is essential to avoid double-counting, unnecessary interventions, and extended follow-up colonoscopies that result from duplicate detections without precise localization.
The authors propose a contrastive learning–based approach using a Siamese neural network architecture.

Specifically:
- The model is trained on the CareForColon2015 trial dataset, which was reconstructed from video sequences by extracting frames that fully capture polyps.
- Multiple ResNet backbones (ResNet18, ResNet34, ResNet50) are evaluated.
- The network is trained with contrastive loss, employing both cosine and Euclidean distance metrics to classify image pairs.

**Strengths:**

1. Appropriate Use of Established deep learning techniques
- The use of a Siamese neural network architecture is standard and well-suited for pairwise matching tasks. It is a sensible choice for polyp matching between image pairs.
2. Clinician-Annotated Dataset / Realistic Data Source
- The dataset is drawn from the CareForColon2015 trial, which is a real clinical study. This gives the data authenticity and relevance.
3. Clarity of the Core Idea
- The flow from problem → dataset → method  → results is coherent.

**Weaknesses:**

## Major Flaws
1. As shown in Figure 2 (lower), Figure 7 (lower), and Figure 11 (lower), the training curves are still decreasing. This suggests that a patience value of 10 may be too small, since it is unclear whether the validation loss might also continue to decrease with longer training.
2. Lines 222–223 state that after selecting a new threshold of 0.15 for the cosine distance, they “retrain the network”. Could the authors clarify why retraining was necessary? Typically, threshold selection is performed during validation without retraining the model, since the threshold only affects decision boundaries at inference.
3. It would have strengthen the paper to have included a comparison against a simpler baseline, such as training the Siamese network with a binary cross-entropy loss.
4. In Figure 1, it would be helpful to add arrows or other visual markers indicating the polyps.
5. A small diagram of the proposed architecture would greatly improve clarity.
6. At line 94, It would be helpful to include a diagram illustrating the five exported images ((labeled 1) through 5)).

## Minor Flaws
1. There are no citations from line 43 until the end of the introduction.
2. The dataset originally contained many more negative polyp pairs, but the authors limited the training set to 5,736 pairs. Given the availability of large numbers of negatives, it would have been natural to explore representation learning methods designed for such scenarios, for example MoCov2 or SimCLR. Could the authors clarify why these approaches were not considered?
3. At line 192, the statement that training took “a few hours” is too vague. Please be more specific.
4. It would be helpful to report the total number of images in the dataset alongside Figure 3 caption.

**Final Justification:**

The work shows potential, and the addition of cross-validation is appreciated. I encourage the authors to strengthen the results and clarify the presentation. However, at present, the lack of stronger quantitative evidence demonstrating the benefit of the contrastive loss, clearer methodological details, and appropriate state-of-the-art comparisons makes it difficult to recommend the paper for acceptance.

**Justification:**

Including the suggested diagrams would significantly improve the clarity and readability of the paper, making it easier for readers to understand both the dataset and the proposed architecture.
The paper has a few major flaws that need to be addressed, I recognize its clinical relevance and the importance of the problem it tackles. I would be open to revisiting my score if these issues are properly addressed.

---

> ### Author Rebuttal · Authors · 2025-10-22
>
> # Major Flaws
>
> ## Decreasing training curves
> Using an early stopping strategy will always contain the risk of not detecting that the validation loss would significantly have decreased at some later epoch number. As a different reviewer have asked for a cross-validation strategy (in stead of us reporting test set results of a single test set/training set-up), we have run a 10-fold cross-validation set-up. Here we have trained the models for 100 epochs. We also check which epoch would have triggered early stopping both for a patience of 10 and 20. In general, it seems like most of the 10 runs do show some small (possible clinically relevant) decrease in validation loss for higher epoch numbers (compared to what we get when using early stopping and a patience of 10). However, it is not given that this relatively small validation set improvement translates into an improvement in test set (and clinical use) performance. That is, one might be overfitting the model, but this one would not know without running different models on the test set. This would again be problematic as one then does not report on test set performances of a true independent test set. These 10 sets of training curves can be added to the appendix in a revised manuscript.
>
> ## Re-training and decision threshold
> This is a point that has also been raised by other reviewers. The intention of our choice was to illustrate that different parameter choices influence the test set performance metrics, but we agree that this was done in a bad way. We will in a revised version of the paper train the networks as before, and then for each trained network run a grid search using the validation set only to produce a threshold value that produce a balanced confusion matrix. This threshold value will then be used to produce the final performance metrics of the untouched test set.
>
> ## Comparison against simpler baseline
> Yes, we could have compared our model against a simpler baseline, such as using a binary cross-entropy (BCE) loss function. This would have answered the question whether the more advanced contrastive loss function results in a better classification model than using the simpler BCE loss. This would be easy to include in a revised version of the paper, as it only entails changing the loss function in the code. However, as other reviewers have pointed out, maybe we should have also compared to relevant state-of-the-art methods. We see it as outside the scope of this paper.
>
> ## Markings of polyps
> This is a very valid comment. As we work daily with these images, it is obvious to us where the polyps are, but this is of course not the case for the general reader. Such markings will be added in a revised version.
>
> ## Diagram of proposed architecture
> As the proposed architecture of a Siamese network with a contrastive loss is very simple and is a standard architecture, we initially chose not to include it. This is easy to add to the paper.
>
> ## Illustrate the five exported images
> At line 94, It would be helpful to include a diagram illustrating the five exported images ((labeled 1) through 5)).
> This we agree would improve the paper. This would also make it clearer why we did not include images “1) First partial” and “5) Last partial” in the training, as one other reviewer asked about. Based on a manual visual inspection of the images, we observed that these partial images often contain just a very small region of the polyp and maybe seen only in the background of the image. These types of images would not be ideal to be considered for a polyp matching approach.
>
> # Minor flaws
>
> ## Lack of citation
> We agree that there could have been more references in the later parts of the Introduction. Much of what is written here is unchallenged “common knowledge” in the colon community, but a few references would have been appropriate to include. This will be included in a revised version.
>
> ## Other representation learning methods
> We acknowledge that methods like MoCov2 and SimCLR are well-suited for scenarios with a large number of negative samples. This is definitely something that can be explored in a future more comprehensive study.
>
> ## Training time
> Yes, the statement “a few hours” of training time is vague. We just wanted to convey that the training time was short and manageable on a normal laptop with a decent graphics card. We have timed a rerun of the training (with a new configuration where we do not use the same patient in more than one set) and the training time was 2 hours and 55 to complete 56 epochs for the Euclidean distance measure of ResNet34. This corresponds to a mean epoch time of 3 minutes and 8 seconds. The other training times can of course be produced if requested. We will include this info in the revised version of the paper. We are in the process of moving the training to run on a server with a high-performance GPU, which will reduce the training time significantly. This is connected to the need for doing a number of 10-fold cross validations.
>
> ## Total number of images of test set
> Yes, we should have included the total number of images of the test set in the caption of Figure 3. This will be done in the revised version of the paper.

---

### Official Review · Reviewer_RiaK · 2025-10-06
**Interesting application, but weak literature review and inconsistent results**

**Rating:** 1
**Confidence:** 4
**Final Rating:** 2
**Final Confidence:** 3

**Summary:**

They study the problem of polyp re-identification from colon capsule endoscopy (CCE), where a pill containing two cameras is swalloed and travels through the intestines. They have developed a sizable dataset of confirmed polyp identities and train siamese resnet models to determine if a pair of images depict the same polyp.
They report accuracies in the range 85-92% and ROC-AUC 96-98%.

**Strengths:**

It is an interesting and relevant cutting-edge problem, and they have developed a large dataset.
The method of siamese resnets is reasonable.

**Weaknesses:**

The literature review is weak. Although polyp re-identification from CCE may not have been studied before, the methodologies from polyp re-identification in colonoscopy is surely relevant, e.g.  https://arxiv.org/html/2502.10054v1.

There are some odd points in the methodology.
They have resized their images to 224x224 to fit the input dimension of ResNets. ResNets are fully convolutional, and can accept any input size. They do not mention pre-training (or random initialization), but even if they want to utilize ImageNet pre-training (which is surely beneficial), this generalizes just fine to different input sizes.
Their handling of decision bias and thresholds is odd, as they appear to re-train the models to achieve this. Why not just adjust the threshold to get a balanced confusion matrix (on the validation set)?

The results appear inconsistent. With ROC-AUC in the 96-98% range, you would expect higher accuracies from a balanced dataset than they report (I haven't completed a proof of this, though, so my intuition may be off).

**Final Justification:**

They were willing to include the exact article I referred to, but this is not enough. A deeper literature review was called for. They were willing to change some of their odd methodological choices, like re-training with different thresholds, but I still think the technical standard is sub-standard. Also their choice to re-scale 256x256 images to 224x224 to fit the original resnet size is very hard to defend.

**Justification:**

In my opinion, the weak literature review and the problematic points in the methodology make this a clear reject, even if the accuracy and AUC results turn out to be compatible.

---

> ### Author Rebuttal · Authors · 2025-10-22
>
> # Literature review
>
> We acknowledge that we should have looked into relevant papers regarding ordinary colonoscopy on reidentification of polyps. However, these two investigation modes are quite different in nature. CCE has lower resolution, i.e., 256x256 pixels as compared to a HD image used in traditional colonoscopy. Capsule camera pills are known to frequently move (large) distances backwards in the intestines. This is not the case for ordinary colonoscopy where the instrument is inserted as far as possible in the colon, and then slowly removed back towards the anus, while the colon is being inspected. Thereby, there will not be any cases of the camera going back and capturing the same polyp at a different time. The paper that you mentioned talks about reidentification on short timescales (seconds and milli-seconds) in ordinary colonoscopy, which is a controlled environment. Furthermore, as the frame rate is high and constant for ordinary colonoscopy, they can use video-based methods. With the highly varying framerate of capsule camera imaging, video-based methods are very difficult to use. However, as mentioned, we should have included this reference and potentially others in our paper.
>
> # ResNet input size
>
> It is correct that ResNets can take many different input sizes. This is also the case when using pre-trained networks but then fine-tuning might take much longer to converge. We realized that we failed to write that the original images are 256x256. With such a small size difference (compared to 224x224), we chose to do the resizing as the loss of detail caused by downsizing should be very small. Running on the full 256x256 will be considered if we set up another experiment with an extensive hyperparameter optimization scheme, which is outside the scope of the proof-of-concept paper.
>
> # Re-training and decision threshold
>
> This is a point that has also been raised by other reviewers. The intention of our choice was to illustrate that different parameter choices influence the test set performance metrics, but we agree that this was done in a bad way. We will in a revised version of the paper train the networks as before, and then for each trained network run a grid search using the validation set only to produce a threshold value that produce a balanced confusion matrix. This threshold value will then be used to produce the final performance metrics of the untouched test set.
>
> # Results – Accuracy and AUC
>
> This comment intuitively feels correct; however, this is not the case. A balanced accuracy of 0.90-0.92 will typically produce these AUCs. A simple simulation trying to separate two 1-D Gaussian classes with means 0.4 and 0.7 (and optimal threshold of 0.5) and standard deviations of 0.15 will produce similar results.

---

### Meta-Review · Area_Chair_ujG5 · 2025-11-01

**Recommendation:** Reject
**Confidence:** 4

**Metareview:**

This submission focuses on a clinically meaningful and somewhat underexplored problem, i.e. determining whether two capsule endoscopy images depict the same polyp. The underlying approach is based on a Siamese neural network and contrastive learning. The motivation and potential long-term impact of the application are clear, and the presentation is generally coherent.

However, the reviewers have raised some concerns around the quantitative analysis and literature, making this paper more like an early version or proof-of-concept than a complete standalone manuscript. Some of the proposed changes would require a revision stage, which unfortunately cannot be accommodated.

While the topic and approach are promising and the authors provided a thoughtful rebuttal, the methodological issues, particularly regarding evaluation protocol, benchmarking, and data handling, mean the paper cannot be accepted in its current form.

---

### Decision · Program_Chairs · 2025-11-05

**Decision:**

Reject

**Comment:**

Based on the reviewers and AC comments, the paper cannot be presented at the conference.